# Position: Prompt Injection Risks Must Be Tackled Interdisciplinarily

## Abstract

As LLM-based systems become pervasive in consumer applications and high-stakes domains, prompt injections (PIs)—the insertion of malicious instructions to manipulate LLM outputs—present a persistent and growing threat. In this position paper, we argue that PIs constitute a socio-technical phenomenon that cannot be addressed by algorithmic defenses alone. Thus, navigating central challenges pertaining to trust and autonomy, responsibility and regulation, as well as innovation and utility requires an interdisciplinary approach integrating technical, legal, ethical, and economic expertise. We here introduce a three-dimensional classification schema to facilitate systematic risk assessment across disciplines, and demonstrate its utility with three case studies ranging from search manipulation to autonomous trading. We call on the research community to embrace interdisciplinary collaboration as an essential part of managing the risks PIs pose to individuals and society.

## 1. Introduction

Despite the enormous potential of LLM-powered AI (Zhang et al., 2024a; Mordechai et al., 2024; Reddy & Shojaee, 2025), there is increasing awareness of associated risks, such as privacy, security, fairness, or accountability attribution (Yao et al., 2024; Gallegos et al., 2024; Das et al., 2024). Although these risks are not unique to LLM-based AI, they are substantially exacerbated by LLMs' vulnerability to so-called *prompt injections* (PIs). In a nutshell, PIs are injections of instructions into the context of an LLM to manipulate its output in order to perform (potentially malicious) actions, such as the generation of harmful content, the performance of unauthorized actions, and the induction of data leakage. Therefore, we dedicate this paper to discussing the potentially disastrous impacts of PIs and how the resulting challenges and risks might be addressed by researchers, developers, deployers, end-users, regulators, and society as a whole. We do so by building on our expertise as an interdisciplinary team of senior scholars in the areas of computer science, cybersecurity, philosophy & ethics, law, and economics who developed a shared position. *Our position is that the risks of PIs can only be appropriately managed when taking an interdisciplinary perspective. This involves a careful analysis of moral and ethical consequences, stakeholders and their responsibilities, and the navigation of trade-offs, such as between opportunities and risks.*

We begin by introducing the problems with PIs (Section 2); we explain what PIs are, why it is difficult to mitigate them, and diagnose them as posing a socio-technical, rather than merely technological, problem. Although this has been previously acknowledged in the literature (Shneiderman, 2020; Selbst et al., 2019), existing work remains to focus on technical mitigation strategies (Liu et al., 2023). But purely technical strategies, we believe, will never successfully address PIs as socio-technical issues. Therefore, we propose a different tack: We argue that dealing with PIs requires an interdisciplinary approach that considers the role of various stakeholders and takes into account technical, legal, ethical, economic, and sociological considerations alike. To facilitate this, we propose a three-dimensional schema for classifying PIs (Section 3). By examining three exemplary cases, we illustrate the various challenges that conceptually different cases of PIs raise. Based on our analysis, we summarize lessons learned and offer concrete recommendations (Section 4). Finally, we discuss alternative views (Section 5) before closing with a conclusion and outlook (Section 6).

## 2. The Trouble with Prompt Injections

Research shows that LLMs are following a sustained scaling trajectory, with performance predictably improving as compute and data increase—evidence that the technology is structurally advancing rather than plateauing (Hoffmann et al., 2022; Henighan et al., 2025). Industry-facing surveys further report that over half—and in some segments more than 70%—of software engineers now integrate LLM tools into their workflows, underscoring real and growing uptake

[1]Anonymous Institution, Anonymous City, Anonymous Region, Anonymous Country. Correspondence to: Anonymous Author <anon.email@domain.com>.

Preliminary work. Under review by the International Conference on Machine Learning (ICML). Do not distribute.

(Zhang et al., 2024b). Complementing this trend, evaluations of agentic AI (Allmendinger et al., 2026) show that autonomous LLM-based agents achieve success rates above 60% on multi-step tasks, suggesting that agentic systems are emerging as a durable extension of the LLM ecosystem rather than a passing novelty (Wang et al., 2024). So in summary, LLMs are becoming integrated into both passive systems and agentic workflows, and even where humans remain in the loop, they increasingly rely on LLM-provided information for decision-making. In such an environment, PIs may face severe consequences.

Against this background, we explain why LLM usage is inherently vulnerable to PIs, and why technological mitigating strategies are crucially limited.

## 2.1. What are Prompt Injections?

Prompt injections (PIs) are injections of malicious instructions into the context of an LLM to manipulate its output in order to perform (potentially malicious) actions. This has rapidly emerged as a critical security and alignment challenge in LLMs, as a recent incident in Google's LLM-based Gemini AI demonstrates (Lakshmanan, 2026). We distinguish two types of PIs: *direct* and *indirect*. In a direct PI, the malicious input is inserted directly into the user input, whereas in an indirect PI, the malicious instructions are embedded in external sources that the LLM processes, such as web pages, documents, or emails. The attacks target either the model, the system, or the user. Goals of those attacks include exfiltrating data (e.g., secret tokens, Personally Identifiable Information, initial system prompts), evading filters (e.g., using in/output encoding), social engineering (e.g., misinformation campaigns, phishing), or performing state-changing actions (e.g., convincing an agent to order something in a web shop).

Initial studies demonstrated that LLMs are susceptible to instruction-following vulnerabilities, in which adversarially crafted prompts—often simple natural language manipulations—can override system directives or bypass safety filters (Schulhoff et al., 2023). Recent works have systematized these attacks, introducing benchmarks (Yi et al., 2025), competitions (HackAPrompt), and attack frameworks that expose the brittle boundary between user intent and model behavior (Samvelyan et al., 2024).

In parallel, the ML security literature has begun addressing defensive strategies, including adversarial training (Mo et al., 2024), certified robustness (Kumar et al., 2024), and behavioral monitoring using attention patterns (Hung et al., 2024). However, these defenses often assume static threat models and struggle against adaptive attackers (Zhan et al., 2025), reflecting an evolving arms race. Furthermore, indirect PIs—where adversarial prompts are embedded in external content—have gained attention for their implications in tool-augmented agents and retrieval-augmented generation (Zhu et al., 2025).

## 2.2. Why mitigating Prompt Injections is hard

The underlying problem of PIs is, by design, hard to defend against. To, for example, summarize an email or a website, the LLM must include the entire document in its context, but within that context, the LLM cannot distinguish between user instructions and textual content. Due to this design, there is no general defense technique against PIs. While previous work has shed light on possible mitigation techniques to tackle the problem, new attacks and bypasses of current defenses are emerging frequently (Chen et al., 2025b). On the other hand, it has been shown that PI detection and prevention methods may degrade performance for benign workloads that do not contain PIs (Liu et al., 2024). Thus, it seems to be difficult, perhaps impossible, to fully defend against PIs on a technical level.

While a naive way to tackle the problem might be to apply input and output filters to LLMs, research (Hackett et al., 2025) has shown that it is trivial to bypass those filters. Also, AI-based pre-filters (Zhang et al., 2025) suffer from the limitation that they can be bypassed, especially by novel, unforeseen, or uncommon attacks. Even if some form of LLM-based filtering may appear to work for, e.g., injected prompts written in English, research in related areas indicates that such filters may fail if the injected prompts are written in a low-resource language, as model performance drops (Agarwal et al., 2024).

Chen et al. (2025a) developed *StruQ*, a system that converts natural-language prompts into structured, validated queries to prevent hidden or malicious instructions from being executed. However, while it reduces the attack surface, attackers can still exploit weaknesses in the schema, identify ambiguities in how natural language is mapped to structured queries, or target parts of the system that remain unstructured to mount attacks. Google presented architectural strategies for AI systems that can be more resistant to PI attacks (Debenedetti et al., 2025) by isolating model inputs, enforcing strict control boundaries, and designing workflows that make it unlikely for untrusted text to influence system-level instructions. A similar approach is implemented by Meta in their "Agents Rule of Two" (Meta AI, 2025). However, in real-world systems, it is hard, if not impossible, to maintain minimal, strict boundaries; even if such boundaries were found, external tools would behave unexpectedly, and developers would rely on the model to self-police instructions rather than enforce hard architectural constraints.

In summary, while technological PI mitigation techniques are important and under active research, we do not foresee them to reliably work on all imaginable PI attacks. We

need additional precautions and a deeper, interdisciplinary analysis of the risks posed by PIs to tackle this problem.

### 2.3. Prompt Injections as a Socio-Technical Phenomenon

As we have seen above, considerable effort has been devoted to mitigating risks associated with PIs through technological means. At the same time, however, scholars in HCI and AI ethics emphasize that PIs actually present a socio-technical phenomenon: they blur authorial intent, shift control from developers to users (or adversaries), and expose deeper challenges related to trust, provenance, and responsibility in AI systems (e.g., Shneiderman, 2020). As such, PIs might also have profound moral and ethical consequences—both for individuals and society at large. Addressing PIs, therefore, requires not only algorithmic defenses but also design principles, usage constraints, and policy frameworks that reflect their interdisciplinary nature (Selbst et al., 2019).

Indeed, PIs pose a general, overarching challenge: when using LLM-based systems, we need to navigate a range of *trade-offs*, e.g., between *economic opportunities* and *fundamental rights*, between individual *security* and *utility*, between *trade secrets* and *disclosure*, or between *innovation* and upholding legal and *moral norms*. While none of these trade-offs are specific to large-scale use of LLM-based systems, the persistent threat of PIs makes it increasingly urgent to address them. The salient trade-offs only become visible once we consider the matter from an interdisciplinary perspective, for we need to take into account not only technological possibilities but also economic opportunities and risks, existing legal provisions and moral ideals, societal desiderata, and different stakeholder interests. To structure the following discussions, we will specifically focus on three sets of issues that each crosscut disciplinary boundaries: Trust & Autonomy, Responsibility & Regulation, and Innovation & Utility.

**Trust & Autonomy.**   PIs expose a fundamental tension between the flexibility that makes these systems useful and the predictability required for trust (the attitude stakeholders have towards a system) and trustworthiness (the system property, (cf. Kästner et al., 2021)): they demonstrate that system behavior can be manipulated in ways neither developers nor regular users may anticipate or control. This raises questions about the standards of robustness against adversarial inputs that should be required for deployment in high-stakes contexts (which relate to regulatory and legal concerns), as well as how to design systems that maintain intended behavior boundaries (which relate to technical concerns). In this context, we might ask what degree of autonomy is adequate for LLM-based systems, to what extent we should ensure human oversight, and how to effectively achieve it.  We might also ask if requiring some kind of

*certification* might be useful to aid system trustworthiness, and if so, in what cases or under which conditions. These questions link ethical and legal considerations about viable norms with those of questions about economic potentials and costs, as well as technological feasibility.

**Responsibility & Regulation.**   When LLM-based systems are manipulated through PIs, we are challenged to attribute responsibility for their consequences.  How should moral and legal responsibility be distributed among the various stakeholders involved? For instance, where should providers be held responsible? What information must be disclosed in what way? Who is *liable* under what conditions? How much technological *literacy* can be expected of average users— and how can this be achieved across a society? What role can an effective regulation of technology play in this context, and what will it look like? Again, these questions need to be addressed in close exchange among technical experts, legal scholars, philosophers, and psychologists.

**Innovation & Utility.**   As we argue above, companies will be pushing the deployment of LLM-based technology in a growing spectrum of applications, including applications for critical and high-risk domains, as they aim to monetize the technology and amortize their enormous investments. Private users will be enjoying the comforts of (agentic) LLM-based systems for delegating dull tasks to save time. In this context, we might not only wonder about the social and economic effects of PIs but also about possible effects on norms and values for individuals and society. For example, we may see that the risks of PIs lead to societal reservations about AI-based innovations or to conservative legislation, which may hinder their widespread adoption. Overall, there are trade-offs between security, regulation, and freedom for technical innovations that can only be answered in an interdisciplinary discourse.

To engage in such a discourse, we propose an approach that will help interdisciplinary groups analyze specific cases of PIs along three conceptually distinct dimensions.

## 3. An Interdisciplinary Approach

To facilitate such an analysis, we propose an interdisciplinary, stakeholder-focused three-dimensional schema for a systematic first-pass analysis and illustrate its utility by examining three exemplary cases.

### 3.1. Three Risk Dimensions: A Classification Schema

We identify three conceptually distinct dimensions to be considered: ***impact***, ***automation***, and ***actors***. These set up a three-dimensional space (see Figure 1). Importantly, each of the three dimensions has several aspects to be considered, and no dimension is tied to any particular discipline; rather,

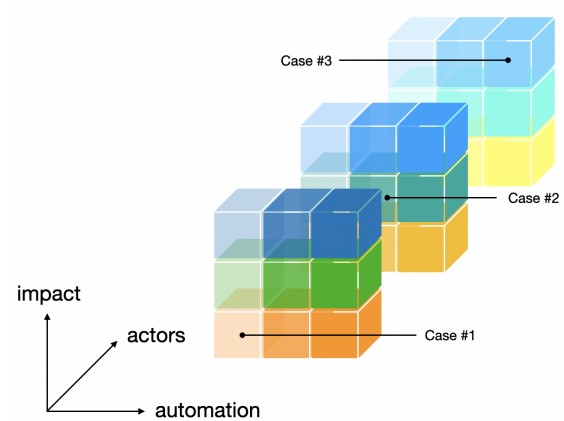

Figure 1. Prompt injection case classification scheme.

classifications along any of the three dimensions cross-cut disciplinary boundaries. We explicate the dimensions as follows:

**Impact** captures the effect size of a PI. This can be specified in terms of *risk* (e.g., how likely is the PI to manipulate behavior) or *stakes* (e.g., how severe or costly will the effects of those manipulations be) or both. An additional consideration under this heading is *long-term vs. short-term* effects. This is particularly relevant where PIs could not only affect a momentary decision but also slowly shift, e.g., cultural norms.

**Automation** captures the degree to which a system or process is removed from human involvement. This may be described as the extent to which there still is a *human-in-the-loop*, or the *degree of control* that humans still have about the system or process in question. Considerations of how *opaque* the system or process in question is to actors might play into this dimension.

**Actors** captures the number and kind of actors involved in a given scenario. This includes *how many* different *actors* are involved and/or affected, as well as what *stakeholder groups* they belong to. In addition, we might be interested in the *intentions* or goals that attackers might have and the role or *status* of their potential *victims*.

Naturally, locating a concrete case along these three dimensions might not be trivial. First, different stakeholders or domain experts might judge cases differently. Second, aspects of **impact**, **automation**, and **actors** may sometimes go hand in hand, potentially resulting in different classifications for a single case. Still, we think it is important to appreciate that these dimensions are, in principle, *conceptually* distinct, even if in practice they sometimes collapse. Theoretically, specific cases of PIs could be located anywhere within the space, the dimensions set up. However, we expect to see clusters in certain areas when analyzing real-world cases.

We offer this schema as a first-pass tool to help interdisciplinary groups analyze PI risks and identify mitigation strategies. Future research will likely refine it, but even in its current form, it serves to differentiate cases and guide targeted responses.

## 3.2. Involved Stakeholders

Before we proceed to our case studies, a note on stakeholders is in order. While the notion of different stakeholders in AI-based ecosystems is familiar (Langer et al., 2021; Tomsett et al., 2018), the exact nomenclature is varied. For the purpose of this paper, we build on the notions utilized by the OECD and the EU AI Act; Figure 2 provides an overview. Accordingly, *"AI models"* designate fundamental components that perform specific tasks, while *"AI systems"* refer to combinations of models with a user interface that provide functional solutions for practical needs. We use the term *"developer"* to describe the entity that develops and trains an AI model. *"Provider"* describes the entity that places an AI model or system on the market or puts it into service for their own purposes. Developer and provider may be identical or different parties. *"Deployer"* refers to parties who use an AI system under their own authority in a professional capacity. Parties who interact with an AI system are designated as *"end users"*; those who are not actually using the system in question but still affected by their operation we call *"affected parties"* and those whose data is being utilized in AI development *"data subjects"*. By *"researcher"* we refer to parties developing, testing, or investigating AI models and systems to gain additional insights about their nature and operation (from different disciplines). When we talk about *"AI agents"* (or agentic AI), we refer to AI systems that can interact with other AI systems (as users), serve as data subjects, injectors and affected parties. Finally, *"injector"* refers to the entity that crafts an injection prompt either directly (e.g., by explicit command) or indirectly (e.g., by sending an email or providing web content) into an AI system. Typically, injectors of direct PIs are end users while indirect injectors are usually data subjects. Our following discussion focuses on indirect PIs since these are considered the most widely used (Paverd, 2025) and the most dangerous (Lakera Team, 2025).

## 3.3. Case Studies

We now discuss three specific cases of indirect PIs. For each case, we briefly outline the technical background, provide a risk assessment based on our three-dimensional schema, and relate it to the interdisciplinary issues outlined in Section 2.3. Based on this analysis, we highlight which challenges must be addressed to tackle PI risks of various kinds.

**Case #1: Search Results.** A website operator equips their website with injected prompts with the aim of convincing an

*Figure 2.* An overview of stakeholders. Note that Researchers and Regulators oversee and investigate all of the different interactions.

LLM to highlight this website as an important source in the LLM's response. If the LLM traverses the website during the retrieval of relevant information to a user query, it will read the injected prompt and may act accordingly.

In this scenario of an indirect PI, the *injector* acts as a *data subject* because the injection is hidden in the information source (the website) that is used by the *AI system*. The *deployer* here is the search engine provider, while *end users* and *affected parties* are all those who use the search engine to retrieve information. Furthermore, (Zou et al., 2025) have shown the effectiveness of injecting malicious data into a RAG system to manipulate LLM responses.

*Risk Assessment:* This case is prototypical for low impact, low automation, and few actors, leading to a low overall risk according to our scheme. The problem is comparable to search engine optimization (SEO), where website owners employ various strategies to improve their visibility in search results. While such practices exist on a spectrum from legitimate optimization to manipulative gaming of algorithms, the impact is limited: it affects specific users who still have to actively click and search through the results, and the effects are typically short-term. At the same time, coordinated PIs may be utilized to spread defamation or disinformation (e.g., in the run-up to an election), which would shift the case toward higher impact.

*Trust & Autonomy:* While individual PIs may not jeopardize system trustworthiness, systematic and continuous PI attacks that manipulate LLM responses for other users may do. Besides, even the threat of PIs may already lead to trust erosion among users. Therefore, it is recommended that developers acknowledge and tackle the problem. Deployers and end users should not blindly trust LLM-based systems but be educated about possible threats and countermeasures.

*Responsibility & Regulation:* We see responsibilities for all

stakeholders. Surely, the injector should be held accountable for injecting misleading information. However, there are also responsibilities assigned to the end users of the LLM: They need to critically assess the trustworthiness of the LLM response, e.g., by verifying information sources and issuing critical follow-up prompts. Finally, system deployers need to install available security measures. Special attention is warranted when PIs lead to systemic societal risks, such as the dissemination of disinformation with the potential to influence the electoral process. Regarding such systemic risks, it may be warranted to impose risk management obligations on the provider of the AI system. It should be noted that regulations such as the EU Digital Services Act already require providers of very large online search engines to implement a lifecycle risk management system (Art. 33 et seq. DSA).

*Innovation & Utility:* Economically, this case resembles a low-cost attention-allocation strategy akin to SEO, where actors invest marginal effort to increase visibility without creating new value, leading primarily to redistribution effects rather than systemic harm (O'Reilly et al., 2024). Such behavior can induce mild arms-race dynamics typical for platform-mediated information markets, but the limited automation and strong user agency constrain large-scale distortions. For platform providers, tolerating some degree of manipulation while applying lightweight countermeasures is often economically efficient, as the marginal cost of perfect filtering may exceed the marginal benefit in user welfare and trust (Hagiu & Wright, 2015).

*Associated Challenges:* The crucial challenge highlighted here is that built-in defenses must be balanced with system utility. In these relatively low-risk cases, the focus might be less on security; vulnerabilities to PIs must be expected and can perhaps best be countered by technological literacy. Responsibility for mitigating PI risks may thus sit more with the end user than developers, providers, and deployers.

**Case #2: Award Selection.** An LLM is used for the initial stage of reviewing academic papers submitted for an internationally renowned research award; it generates a shortlist that is then further reviewed by a human committee. PIs by the authors could lead the system to shortlist their papers or exclude papers from rivaling groups.

In this scenario of an indirect PI, the *injector* acts as a *data subject* as the data (the application) that is processed by the *AI system* contains the PI. The *deployer* offers the AI review service, and the *end-user* is the organization that generates the shortlist. Notably, the set of *affected parties* at least includes all people who applied for this research award (including the *injector*), as well as the decision committee. Research by (Lin et al., 2025) shows that PIs can severely mislead LLM-based reviewing of academic papers.

*Risk Assessment:* This case stands as a prototype for a medium impact, medium degree of automation, and a medium number of actors. In terms of impact, the award is prestigious and may affect the careers of both the awardees and those who do not receive it as a result of the PI. There is some level of automation, as the shortlist is generated by the LLM. Still, the final decision over the awardees remains in the hands of human actors. Finally, the affected parties do not only comprise the injectors who may receive the award, but also the larger scientific community.

*Trust & Autonomy:* This case represents a distortion of a high-stakes tournament-style allocation mechanism, where scarce reputational rewards (prestige, signaling value, career capital) are distributed among competitors (Connelly et al., 2014). PIs introduce asymmetric manipulation opportunities that undermine allocative efficiency by favoring strategic behavior over scholarly quality, potentially leading to misallocation of reputational rents and reduced trust in the award as a signaling device (Frey & Gallus, 2017). As a consequence, available technical counter-measures (such as filtering and validation) must be enforced, and given their imperfection, it is not advisable to completely trust the output of an LLM.

*Responsibility & Regulation:* This case raises a number of interesting questions about responsibility attribution and how to navigate the trade-off between automation (compensating for lack of human labor) and satisfying central societal desiderata (like fairness, upholding norms for good scientific practice, etc.). From a technical perspective, it is not enough to scrutinize the output of the model (*Are the shortlisted papers eligible for the award?*), as the PI may lead to filtering out eligible papers that may have been selected in a manual process. From a regulatory perspective, there is no need for legislative intervention: Entities presenting the award may forbid the use of PIs in their terms and conditions. Such terms and conditions will withstand the test of unfair terms legislation.

*Innovation & Utility:* Unlike low-impact visibility manipulation, such distortions generate negative externalities for the broader scientific community by weakening incentives to invest in genuine research quality. As a result, the expected social cost is higher, justifying stronger safeguards despite increased implementation costs.

*Associated Challenges:* This case illustrates that to mitigate PI risks, full automation should already be avoided in medium-risk cases. Beyond that, measures similar to those discussed in Case #1 might be taken (especially end-user literacy). Additionally, we think that in medium-risk cases responsibility is shared between more actors, albeit still primarily end users. Clear guidelines on AI use for different circumstances should be formulated by user communities (e.g. funding organizations, research communities,

companies, etc.) and enforced by penalties.

**Case #3: Stock Exchange.** Various agentic AI systems collaborate to perform actions on the stock exchange. They can sell and buy stocks in the name of corporations or banks, and there is no longer any active human confirmation for individual transactions. In this scenario, agents are vulnerable to PIs hidden in, e.g., documents they retrieve or answers they receive from tool calls. Accordingly, PIs could modify the behavior of AI agents, causing financial harm to stakeholders.

The *deployers* in this scenario are those parties that instantiate the (different) agents. Given that no humans are directly involved here, the *injector* again acts as a *data subject* as the documents, news articles, tools, etc., are processed by the *AI system*, which can contain a PI. Also, we do not have an *end user* in this scenario as there is no human interaction. The set of *affected parties* covers a plethora of entities, from the stocks that are traded, to the entities that are trading, up to essentially all people that rely on a global market and economy. (Debenedetti et al., 2024) show that agentic frameworks are inherently vulnerable to PI attacks and that none of the existing defenses can completely prevent them.

*Risk Assessment:* This case represents a high-risk scenario, combining a high impact of the PIs with a high degree of automation and a large number of affected actors. The consequences of a harmful PIs could be catastrophic, potentially leading to large disruptions of global financial markets affecting millions of people; cascading effects may lead to further disruptions in other systems.

*Trust & Autonomy:* This case constitutes a systemic-risk scenario rather than a localized incentive distortion, as autonomous trading agents operate in tightly coupled markets where errors can rapidly propagate (Perrow, 1984). PIs that alter agent behavior can trigger negative externalities and cascading failures, similar to those studied in the literature on algorithmic and high-frequency trading, where small perturbations can be amplified through feedback loops and liquidity dynamics (Kirilenko et al., 2017). In such a setting, it may be difficult to fully trust automated agentic systems; safeguards and the possibility of manual intervention need to be in place.

*Responsibility & Regulation:* This case may be subject to already existing risk management obligations in the financial sector, which are country-specific (Lee & Schu, 2022). Generally speaking, we see obligations of the provider and the deployer to perform appropriate risk management and prevent failures that can have catastrophic consequences. Regarding the injector, general anti-fraud regulations apply. Regulations should address both: risk management ex ante (because public interests are concerned) and liability rules ex post. Losses will be scattered among a large number

of parties, thus, effective class or representative actions are required. It is the responsibility of the deployer to apply technical counter-measures such as checks for task alignment (Jia et al., 2025) to minimize technical risks.

*Innovation & Utility:* Because losses are not confined to the manipulating actor but are borne by a wide set of market participants, the private incentives to secure individual agents may fall short of the socially optimal level, justifying strong ex ante regulation and centralized safeguards. In contrast to Cases #1 and #2, market trust and financial stability themselves become public goods at risk. Hence, it is advisable to adopt a more conservative approach and accept the limitations of LLM utility to enhance system security.

*Associated Challenges:* This case highlights that regulation is in order where risks are high and the effects of PIs can be potentially devastating to society (possibly on a global level). A second challenge this case highlights is that where no end users are present, responsibility sits with deployers; they could be required to implement pre-deployment controls (e.g., access to only approved data sources), stress tests, sandboxing, safeguards (caps), or kill switches. A major challenge with AI-based algorithmic trading is the unpredictability of algorithmic decisions (Lee & Schu, 2022); PIs exacerbate the problem and should be considered. Generally, regulators should ensure that an appropriate framework is set specifying such conditions, possibly varying with the severity of associated risks.

## 4. Lessons Learned

Our interdisciplinary analysis of the three use cases, utilizing our stakeholder taxonomy and risk-classification schema, has revealed that there is no one-size-fits-all solution to manage PI risks. Generally speaking, we found that, since developers and providers can most likely not fully guarantee technical mitigation of PI risks, there is a *de facto* increased responsibility for end users and deployers, even if moral and legal standards might suggest otherwise. This is precisely why PIs must be tackled interdisciplinarily. Besides, our case studies have highlighted that both possible mitigation strategies *and* responsibilities for actively addressing PI risks may shift systematically with risk levels. Thus, we suggest that different stakeholders need to pursue distinct mitigation strategies in low-, medium-, and high-risk cases. Below, we detail some concrete recommendations.

**For End Users** we suggest primarily focusing on technological literacy, education, and public awareness. We specifically recommend AI training at the workplace, and welcome critical reports about AI (Doss et al., 2025). With increasing risks, end users should be skeptical of (full) automation, establish policies for good AI use in their communities (e.g., funding organizations, research communities,

companies, etc.), and enforce these with penalties. For high-risk cases where community commitment does not seem enough, *certification* could help promote safe AI use across society (Durán, 2026).

**For Developers & Providers** we submit that they owe a duty of care to minimize potential harms to end users, data subjects, and affected parties. Thus, we urge that they must acknowledge (i.e., be transparent about) and tackle vulnerabilities to PIs, e.g., by training LLMs to recognize PIs or employing output filtering and validation (e.g., by checking links); to this end, techniques from web spam detection may be applicable (Ntoulas et al., 2006). For medium-risk scenarios, we additionally propose implementing pre-deployment controls (e.g., access restrictions). Where risks are high, we recommend sandboxing, subjecting AI systems to rigorous stress tests, and implementing safeguards (e.g., kill switches). Also, Developers & Providers should always keep an eye on new mitigation techniques and see if they can be applied to their system.

**For Deployers,** our recommendations are somewhat similar to those of developers: the best possible effort is required to mitigate PI risks, even if this involves combining many different imperfect strategies. Beyond that, deployers must bear in mind both system usability and end-user needs; they should generally be transparent about both system vulnerabilities and how these are addressed. With increasing risks, (full) automation should be treated with care. For agentic systems, enforcing strict boundaries between different agents may be beneficial, such that each agent only has the minimum access rights required to perform its individual job (Debenedetti et al., 2025) (say, access to email *or* calendar entries). For high-risk cases specifically, safeguards (e.g., caps or kill switches) must be made easily usable. Similar to the Developers & Providers, Deployers should keep themselves up-to-date with new mitigation techniques.

**For Regulators,** we posit that PIs are not unlawful *per se*, but may be used as an instrument for illegal purposes. Still, AI-specific regulations can require *ex ante* risk mitigation, as envisaged, e.g., by the EU AI Act. To this end, regulatory requirements should be calibrated to the risk profiles of system deployment. In relatively low-risk cases, general provisions (e.g., rules of fraud) will usually be sufficient to address harms caused by PIs. As risks increase, stringent transparency requirements might be imposed (as exemplified by the EU AI Act) in addition to requiring developers and providers to combat PI risks with available measures. Besides, full automation should be clearly prohibited in high-risk cases, and implementing safeguards should be mandatory. Currently, both the EU and the US have designed safe havens for intermediary service providers (sec. 230 US Communications Decency Act, Art. 4 et seq. EU

Digital Services Act). It is our position that LLM providers do not and *should not* benefit from a liability exemption under these rules: LLMs create their own content, and they do not solely mediate third-party content. Providers and/or deployers bear responsibility for this LLM-generated content, including output skewed by PIs.

**To Researchers,** we call to tackle PI risks interdisciplinarily, bearing in mind different stakeholders, risks, and tradeoffs to be navigated. Generally, we hope that researchers will communicate their insights about system vulnerabilities to the public to help users protect themselves. Besides, researchers may investigate strategies for effective self-regulation within communities, assess the extent to which global regulatory efforts could be feasible, clarify risk profiles and different cases, illuminate which safeguards pre-deployment controls are most appropriate in specific circumstances, and develop reliable certification protocols. Finally, we think it is worth asking whether there are (ethically, legally, morally) permissible PIs—for instance, injections deployed by law enforcement during a criminal investigation to intercept or redirect a malicious agent. And if this is so, who is authorized to decide about them, employ them, and where and how to draw the line to impermissible PIs?

## 5. Alternative Views

Our discussion thus far has shown that an interdisciplinary treatment of PIs is vital to understanding, and, down the line, effectively mitigating the many different risks associated with them—both for society and individuals. Before we close, we shall briefly comment on two alternative views that suggest we might get by without these interdisciplinary efforts.

First, consider the *techno-optimist position*. According to this view, PIs generally are just a temporary risk and will be fully mitigated once LLMs become more capable in detecting and avoiding them. Generally, LLMs are taken to have a favorable impact on society, and we need not fear they pose any existential risks (e.g., König, 2022). Thus, research on PI risks and risk mitigation may be considered as a waste of time and resources; concerns about *trust and safety* or *risk management* as "enemies" of progress (e.g., Andreessen, 2023).

We think this is overly optimistic, to say the least. While we agree that research on how to immunize LMMs against PIs will be needed, we doubt that this alone will be sufficient. Even if successful in the end, developing the required tools takes time, while PIs pose *immediate* risks. For another, our analysis has clearly demonstrated that a lot of ethical, legal, and economic questions need to be addressed for effective PI risk mitigation—and that will likely never be achieved through technological development alone.

Second, consider the *techno-pessimist position*. According to this view, the inherent vulnerability of LLMs to PIs should lead us to heavily regulate, or even prohibit, LLM usage (at least in critical or 'high-risk' domains). Some even go as far as to predict that rapid technological developments will yield the extinction of humanity (e.g., Kokotajlo et al., 2025) unless strict regulation is put into place.

While we agree that the pace of the current technological developments contributes to the urgency of dealing with PI risks, we do think that an overly strict regulatory approach is throwing out the baby with the bathwater; for major prohibitions against LLM-based systems are not only unrealistic in a globalized world but also morally questionable, likely economically unwise, and hardly enforceable.

Still, we submit, leaving technology developers completely free rein and turning a blind eye to possible harms that PIs might cause is unwise. What is needed, thus, is a reasonable middle way that helps us effectively navigate the many trade-offs we discussed above—for researchers, users, deployers, regulators, and society at large. We believe that our interdisciplinary approach, analyzing specific cases along several dimensions, will help pave the way towards such a nuanced treatment of LLM-based systems.

## 6. Conclusions

Dealing with PIs requires walking a tightrope: we need enough control over LLM-based technology to minimize risks, especially in critical domains, while still leaving enough leeway for progress. We firmly believe that this requires an interdisciplinary effort. Analyzing the problem from a single perspective—be it technical, economical, ethical, or regulatory—runs the risk of leading into overly optimistic (*"We can prevent prompt injections completely."*) or pessimistic (*"We need to legally prohibit LLM usage for automated decision making."*) views that are unlikely to find broad adoption. Instead, we take it, it is time to address the risks associated with PIs in a balanced manner. We suggest that involving experts from different fields to recognize the roles of different stakeholders, distinguish different risk categories, and appreciate the various challenges and tradeoffs associated with PIs is crucial to mitigating PI risks. To this end, we proposed a clear stakeholder taxonomy and a three-dimensional risk-classification schema. For the purposes of this position paper, our discussion had to be limited to three relatively simple example cases falling into different categories along each of the dimensions. However, reality is much more complex. Future research will thus have to apply the schema to real-world cases and refine it further.

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
