# OpenReview forum: "Position: Prompt Injection Risks Must Be Tackled Interdisciplinarily"
_ICML.cc/2026/Position_Paper_Track — Submitted to ICML 2026 Position Paper Track_

### Official Review · Reviewer_qVfV · 2026-03-05

**Significance:** 3
**Argument Clarity:** 3
**Rating:** 4
**Confidence:** 3

**Questions:**

Please refer to Weaknesses.

**Alternative Views Section:**

Yes

**Compliance With Llm Reviewing Policy A Conservative:**

Affirmed.

**Discussion Potential:**

3

**Paper Summary:**

This position paper addresses a critical and escalating threat in LLM-based systems: prompt injections (PIs), where malicious instructions manipulate model outputs for harmful purposes. The core argument is that PIs are not merely a technical vulnerability but a socio-technical phenomenon, requiring integration of technical, legal, ethical, and economic expertise to mitigate effectively. Overall, an important concept addressed by the manuscript is the limitations of purely algorithmic defenses against PIs—given the inherent design of LLMs to process contextual content without distinguishing user intent from embedded instructions. The authors propose a three-dimensional classification schema (impact, automation, actors) to facilitate systematic interdisciplinary risk assessment and illustrate its utility through three case studies spanning low (search result manipulation), medium (academic award selection), and high (autonomous stock trading) risk scenarios. Overall, the authors assess a central aspect: how stakeholder responsibilities (end users, developers, deployers, regulators, researchers) and mitigation strategies shift with risk levels, emphasizing that no one-size-fits-all solution exists. The work engages with alternative techno-optimist and techno-pessimist views, advocating for a balanced, interdisciplinary approach that navigates trade-offs between innovation, utility, trust, and security.

**Position:**

Yes

**Position In Title:**

Yes

**Related Work:**

3

**Strengths And Weaknesses:**

Strengths

1. The paper breaks new ground by framing PIs as socio-technical issues rather than purely technical ones. This perspective addresses a major limitation of existing literature, which overemphasizes algorithmic defenses while neglecting ethical, legal, and economic dimensions—an essential shift given the pervasiveness of LLMs in consumer and high-stakes domains.
2. The three-dimensional (impact-automation-actors) schema and stakeholder taxonomy provide actionable tools for cross-disciplinary teams to analyze PI risks. Unlike abstract frameworks, these tools are grounded in real-world case studies, making them useful for researchers, practitioners, and regulators alike.
3. The three case studies (search manipulation, award selection, autonomous trading) are carefully chosen to span risk spectrums, enabling the authors to derive nuanced recommendations for different scenarios. Each case analysis integrates technical feasibility, legal obligations, ethical considerations, and economic incentives, demonstrating the framework’s holistic value.
4. The paper thoughtfully addresses techno-optimist (PIs are temporary) and techno-pessimist (LLMs should be prohibited) views, avoiding extreme positions. This balanced approach enhances credibility and broadens the paper’s appeal to diverse stakeholders, from industry developers to policymakers.
5. The manuscript provides concrete, differentiated guidance for end users (technological literacy), developers (duty of care, pre-deployment controls), deployers (sandboxing, kill switches), regulators (risk-calibrated rules), and researchers (interdisciplinary collaboration). These recommendations are practical and aligned with existing legal frameworks (e.g., EU AI Act, DSA), ensuring real-world applicability.

Weaknesses

1. While the case studies illustrate the schema’s utility, the paper lacks empirical evidence of its effectiveness—e.g., surveys of interdisciplinary teams using the framework, or quantitative assessments of how well it predicts PI risk severity. Such validation would strengthen the case for widespread adoption.
2. The legal analysis focuses primarily on EU and US frameworks, with minimal attention to cross-cultural differences in privacy norms, liability rules, or regulatory approaches (e.g., Asia, Africa, Latin America). This limits the framework’s generalizability to global LLM deployments.
3. The classification schema defines "low/medium/high" risk along the three dimensions but provides no clear thresholds (e.g., how to quantify "high impact" vs. "medium impact"). This ambiguity could lead to inconsistent application across different use cases or stakeholder groups.
4. The paper calls for interdisciplinary collaboration but offers little practical guidance on how to operationalize it—e.g., how to resolve conflicts between technical feasibility and legal requirements, or how to structure teams with diverse expertise to address PI risks effectively.
5. The analysis focuses on indirect PIs (the most prevalent type) but overlooks edge cases such as cross-lingual PIs, PIs in multi-modal LLM systems (e.g., image-text models), or PIs targeting LLM agents interacting with IoT devices. These edge cases present unique socio-technical challenges that the current framework does not address.

**Support:**

3

---

> ### Author Rebuttal · Authors · 2026-03-30
>
> We thank Reviewer qVfV for their appreciation of our 3D schema. Below we address their worries by specifying our position.
>
> **W1: Empirical evidence.** *(See also Review hN3U Q1.)*
> We developed the schema as an interdisciplinary group of scholars in CS, cybersecurity, philosophy & ethics, law, and economics---precisely the kind of team we advocate for. The case analyses in Section 3.3 reflect our joint deliberation process. This framework is intended to serve as the foundation for empirical work across a wide range of real-world use cases from the community, which will not only provide external validation of the schema itself but also address gaps, such as the need for more reference cases. Broad validation with a wide variety of real-world use cases and large-scale survey studies will require community involvement. This cannot be delivered by a position paper; but we aim to kickstart such initiatives with our contribution.
>
> **W2: Focus on EU and US frameworks.**
> We acknowledge that there are wide differences in regulatory approaches across countries. An interdisciplinary paper can only scratch the surface of the complex national and global regulatory framework. The paper addresses that risk-management obligations and general anti-fraud regulations are country-specific (p. 6). We furthermore point to specific legislation which we believe to be particularly influential in the global context. In particular: Sec. 230 US Communications Decency Act has laid the ground for the online intermediary services market as it exists today. This rule has been copied by many nations around the world. The EU’s AI Act, on the other hand, is the first comprehensive regulation on AI by a major regulator in the world. European legislation has been shown to often spread beyond the EU’s borders as a consequence of the so-called Brussels effect. This effect may arise due to (1) regulated entities complying with EU laws even outside the EU for a variety of reasons or to (2) regulators in other parts of the world modelling their own, adapted and often improved rules upon EU legislation. We therefore believe it is appropriate to address these specific pieces of legislation.
>
> *Anu Bradford, The Brussels Effect, 107 Nw. U. L. Rev. 1 (2012).*
>
> **W3: Risk quantification.** *(See also Review hN3U W1).*
> We appreciate this observation and want to clarify a deliberate design choice. Our schema is intentionally qualitative and intended as a first-pass structuring tool for interdisciplinary groups---not (yet) as a quantitative risk-scoring instrument. We argue this is the appropriate level of formalization for a position paper that targets researchers across disciplines (CS, law, ethics, economics), many of whom would not share a common formal language for risk quantification. Imposing numerical scales prematurely would risk false precision and obscure the very tensions we aim to surface. That said, we fully agree that future empirical work should develop operationalized versions of the schema. We note that established frameworks such as the NIST AI RMF similarly begin with qualitative risk tiers before operationalizing them---our schema occupies an analogous early-stage role, specifically tailored to the PI threat landscape. We will make this clear in a revised version of the paper.
>
> **W4: Interdisciplinary Collaboration in Practice.** We developed our schema by bringing together experts from CS, cybersecurity, philosophy & ethics, law, and economics. Precisely such an interdisciplinary consortium will be needed to effectively address PI risks (as well as other LLM-safety risks) by overseeing the many relevant factors and dimensions. Besides, in any real-world case, tradeoffs between technical, legal and ethical concerns need to be negotiated relative to their specific context. There is, therefore, no one-size-fits all solution, or any one precise algorithm to determine how experts from different domains will achieve a consensus view. Precisely because there are so many dimensions and contextual factors to be considered, our 3D schema is so highly valuable: It provides guardrails as well as a clear conceptual starting point for analyzing a given case so experts can start their discussion from a common ground.
>
> **W5: Other types of PIs.** Indirect PIs as an interpretation of malicious instructions embedded in external and untrusted inputs are independent of language, modality, and system architecture. The cases of cross-lingual content, multi-modal inputs (e.g., images, PDFs, voice, etc), and IoT data streams (e.g. cameras) simply represent different input channels through which those indirect prompt injections can occur. Thus, they do not constitute separate categories as the underlying problems caused by those injections are still the same independent of the input channel. We acknowledge to discuss various input channels in more breadth and extend our general definition of PIs to make the point about delivery channels clearer.

---

> > ### Author Rebuttal · Reviewer_qVfV · 2026-04-06
> >
> > The authors addressed most of my concerns, so I keep the original positive rating.

---

### Official Review · Reviewer_XJng · 2026-03-07

**Significance:** 3
**Argument Clarity:** 2
**Rating:** 4
**Confidence:** 3

**Questions:**

- Can the authors better argue why Case #1 is of "low impact, low automation and few actors"?
- Why not make the paper about "AI security" more generally than just PIs?
- What concrete measures could ICML or AI research/regulation institutions consider, as a response to the paper's position?

**Alternative Views Section:**

Yes

**Compliance With Llm Reviewing Policy A Conservative:**

Affirmed.

**Discussion Potential:**

2

**Paper Summary:**

The paper argues that mitigating prompt injections (PIs) should be an interdisciplinary effort, as it involves social science, legal and philosophical dimensions such as agency, responsibility, utility and risk evaluations. The paper especially highlights analyzing PIs based on impact, automation and actors. Such analyses are provided for three use cases, to showcase their values. It then provides calls to action for different communities, as well as responses to alternative views.

**Position:**

Yes

**Position In Title:**

Yes

**Related Work:**

3

**Strengths And Weaknesses:**

Strengths:
- The paper tackles a very important current and future issue for LLM-based AI deployments.
- It correctly highlights the need to take a broader perspective than the technical one, by involving social scientists and philosophers.

Weaknesses:
- Section 2.2 lacks a discussion on sandboxing. This has become a leading technical solution to mitigate risks, by simply putting boundaries on read/write access of an agentic AI. I believe that many of the challenges put forth (responsability, autonomy, risk) are further highlighted by the choice of default sandboxing setting. Concretely, Anthropic Claude Code's default has "Default read behavior: Read access to the entire computer, except certain denied directories".
https://code.claude.com/docs/en/sandboxing
- Section 2.3 on Responsability. I believe that the responsibility of managers who are pushing for AI adoption should also be considered.
- Section 3.1. "Human-in-the-loop" has also been criticized, e.g. https://philpapers.org/rec/TAYWHA-3. Issues may include job satisfaction, approval fatigue, AI sycophancy. In particular, it could be argued that liability should not rest solely upon the human in the loop.
- I found the classification of Case #1 (Search Results) as "low impact, low automation and few actors" deeply disturbing. The SEO optimization market is estimated at 100 billion USD, which is 5 times more than OpenAI's 2025 revenues. State actors are known to produce massive amounts of misleading information online, including astroturfing, including the Pravda Network targeting LLMs, which has already led to the cancellation of a Presidential election in a NATO country (Romania).
- Line 380, left column. I would add "AI security training". Companies are already full of "AI training", which is mostly about basic prompting, and does not protect against PIs.
- I am not sure I see the value of restricting the paper to PIs. The arguments seem to apply more generally to AI security, if not AI liabilities.
- I feel that the paper could give more room "to inspire constructive, useful discussion within the ICML community", by providing more open and concrete questions on AI deployment and AI research. Should there be a new track dedicated to interdisciplinary responses to AI security? Should there be a systematic highlight of PI vulnerabilities and mitigations for ICML submissions of new algorithms?

**Support:**

2

---

> ### Author Rebuttal · Authors · 2026-03-30
>
> We thank Reviewer XJng for their appreciation of our interdisciplinary perspective as well as for the constructive suggestions for improvement.
>
> **W1: Sandboxing.** Sandboxing can be categorized as an architectural strategy, as we discuss some selected representatives in Section 2.2. We also explicitly refer to sandboxing in our recommendations for developers and providers in Section 4. Sandboxing can be used in an attempt to limit the harm of PIs in agentic systems (which may lead to trade-offs with utility; see the current discussion around whether OpenClaw should have access to your mailbox), but it would not solve all PI risks. For example, in our case studies #1 and #2, we do not immediately see how sandboxing could help. However, we will explicitly mention sandboxing as a mitigation technique in Section 2.2 in the revised paper.
>
> **W2: Responsibility.** Good point! We already touch upon this aspect in our recommendations for “end users” in Section 4 (*“...end users should be skeptical of (full) automation…”*), understanding said managers as end users even if they do not directly interact with the LLM system. However, we acknowledge to explicitly discuss the broader responsibility of “deciders” in Section 2.3.
>
> **W3: Human in the loop.** We mention the human-in-the-loop in Section 3.1 merely as one proxy that might (along with opacity and degree of control) help determine where a given case is to be located on the automation scale; we neither defend that it constitutes full control, nor that it is uncritical to keep humans in the loop – this is a discussion well beyond the scope of our paper.
>
> **W4: Classification of Use Case #1.** The classification depends on the scale and the intended impact of the prompt injection (see our risk schema). We do not intend to state that all prompt injections that manipulate search results have a low risk. In our analysis, we explicitly state that *“...at the same time, coordinated PIs may be utilized to spread defamation or disinformation (e.g., in the run-up to an election), which would shift the case toward higher impact.”* This discussion is exactly why we need the interdisciplinary view and the risk assessment scheme proposed in this paper!
>
> **W5: AI Security Training.** Good catch! We will add “AI security” training and differentiate it from general “AI training”.
>
> **W6: Generalization beyond PIs.** *(See also Review hN3U Q2.)*
> In its current, relatively abstract form, we indeed do think that there is potential for such a generalization, though our focus is on prompt injections. We picked this focus deliberately to do justice to the limited length of a position paper. But it is clearly true that other types of LLM safety risks, such as jailbreaks and hallucinations, also elude mere technical mitigations and would thus benefit from a multidimensional interdisciplinary analysis. If the dimensions we suggest for PIs will transfer directly or if they might have to be adapted or just explicated with different concrete scales, remains to be investigated when studying other LLM risks in more detail.
>
> **W7: Concrete recommendations for the ICML community.**
> We thank the reviewer for this suggestion---we agree that our paper should more directly address what the ICML community can do. In the revised manuscript, we will expand Section 6 (Conclusions) with concrete, actionable questions and proposals directed at the ML research community. These include: (1) whether ML venues should require structured PI robustness assessments for submissions that propose new LLM-based systems or agents, analogous to broader impact statements; (2) whether dedicated interdisciplinary tracks or workshops at ML conferences could facilitate the collaboration we advocate; (3) whether shared benchmarks and red-teaming protocols for PI resilience should become standard evaluation practice; and (4) how ML researchers can systematically engage legal, ethical, and economic expertise in their work---e.g., through interdisciplinary co-authorship incentives or structured review criteria. We believe these additions will make our position more actionable for the ICML audience specifically.
>
> **Q1:** see W4
>
> **Q2:** see W6
>
> **Q3:** see W7

---

> > ### Author Rebuttal · Reviewer_XJng · 2026-04-05
> >
> > I thank the authors for their rebuttal. I'd like to come back to W1 and W4.
> >
> > W1. Sandboxing indeed does not help much to Case #1 and Case #2. However I believe that it is essential for agentic AIs, which can undertake irreversible actions under PI attack. As there is a growing call to use such AIs, I believe that sandboxing is important to highlight (which I trust the authors will).
> >
> > W4. I am still deeply disturbed that Case #1 could be viewed as lower impact than Case #2, especially given the current large-scale information manipulation context. I strongly disagree with the sentence "This case is prototypical for low impact, low automation, and few actors, leading to a low overall risk according to our scheme". Coordinated AI-powered propaganda is arguably huge impact (including the destabilization of democracies, see https://www.nature.com/articles/s41586-026-10098-2), high automation (see e.g. this journalistic investigation https://forbiddenstories.org/team-jorge-disinformation/) and a huge number of actors (this French report estimates that 20M chinese individuals are paid to produce disinformation, https://www.irsem.fr/rapport.html). The stakes have grown significantly over the last few months, as both the US and EU are investigating the security of search/recommendation systems (https://www.nytimes.com/2026/03/25/technology/social-media-trial-verdict.html, https://www.reuters.com/legal/litigation/dutch-court-upholds-ruling-forcing-meta-offer-chronological-feeds-2026-03-10/). At the very least, the authors must provide a significantly improved justification of their claim.

---

### Official Review · Reviewer_K7zA · 2026-03-13

**Significance:** 2
**Argument Clarity:** 3
**Rating:** 4
**Confidence:** 3

**Questions:**

See weaknesses.

**Alternative Views Section:**

Yes

**Compliance With Llm Reviewing Policy A Conservative:**

Affirmed.

**Discussion Potential:**

2

**Final Justification:**

The reviewer holds their original evaluation (weak accept) for this paper.

**Paper Summary:**

This paper proposes addressing PI risks from an interdisciplinary perspective. The paper proposes three dimensions for evaluating PI risks and then conducts three case studies using these dimensions, highlighting the associated risks.

**Position:**

Yes

**Position In Title:**

Yes

**Related Work:**

3

**Strengths And Weaknesses:**

### Strengths:
- Three case studies in this paper play a key role in substantiating the importance of PI risks.
- Enough preliminary information of the PI is provided, which is helpful for readers to access the content.

### Weaknesses:
- It would be better to provide more quantitative metrics so one can easily evaluate the risks. That said, the reviewer understands the difficulty of doing so in a general framework.

**Support:**

3

---

> ### Author Rebuttal · Authors · 2026-03-30
>
> We thank Reviewer K7zA for acknowledging the key role of our case studies and highlighting the accessibility of our manuscript.
>
> **W1: Quantitative metrics.** *(See also Review hN3U W1.)*
> We appreciate this observation and want to clarify a deliberate design choice. Our schema is intentionally qualitative and intended as a first-pass structuring tool for interdisciplinary groups---not as a quantitative risk-scoring instrument. We argue this is the appropriate level of formalization for a position paper that targets researchers across disciplines (CS, law, ethics, economics), many of whom would not share a common formal language for risk quantification. Imposing numerical scales prematurely would risk false precision and obscure the very tensions we aim to surface. That said, we fully agree that future empirical work should develop operationalized versions of the schema. We note that established frameworks such as the NIST AI RMF similarly begin with qualitative risk tiers before operationalizing them---our schema occupies an analogous early-stage role, specifically tailored to the PI threat landscape. We will make this clear in a revised version of the paper.

---

> > ### Author Rebuttal · Reviewer_K7zA · 2026-04-02
> >
> > The reviewer thanks the authors for their responses, and will main their evaluation on this paper.

---

### Official Review · Reviewer_hN3U · 2026-03-18

**Significance:** 3
**Argument Clarity:** 3
**Rating:** 4
**Confidence:** 2

**Questions:**

- The schema validation. Have the author tested the three-dimensional schema with actual interdisciplinary groups like non-ML experts?
- Can the three-dimensional schema generalize to the other LLM safety risks (jailbreaks, hallucinations, etc)?
- Can the framework adapt to the injectors who also adapt to governance measures?

**Alternative Views Section:**

Yes

**Compliance With Llm Reviewing Policy A Conservative:**

Affirmed.

**Discussion Potential:**

3

**Final Justification:**

Sorry for the late reply. All the concerns are addressed. I lean to accept this paper. For the strength, I do find it hard to distinguish the boundary between position paper and normal paper because there is no experimental results here, making me hard to improve the strength normally. The idea itself is great.

**Paper Summary:**

This paper presents that prompt injections represents a socio-technical phenomenon that inherently escapes purely algorithmic management. Technical countermeasures degrade benign performance and cannot address legal ethical and economic questions. The contributions of this paper are:

- a three-dimensional risk classification schema
- a stakeholder taxonomy adapted from OECD/EU AI Act.
- three case studies across risk levels

**Position:**

Yes

**Position In Title:**

Yes

**Related Work:**

3

**Strengths And Weaknesses:**

Strengths:

- The position is clear.
- The interdisciplinary is impressive.
- The reference is complete and latest.

Weaknesses:

- Schema is informal. There is no measurement scales, aggregation rules or conflict-resolution procedures.
- The interdisciplinary call is not that novel. [1, 2] is under-explored.
- Limited engagement with AI governance academic literature, like NIST AI RMF, AI liability scholarship, etc.

[1] Selbst, Andrew D., et al. "Fairness and abstraction in sociotechnical systems." Proceedings of the conference on fairness, accountability, and transparency. 2019.

[2] Shneiderman, Ben. "Human-centered artificial intelligence: Reliable, safe & trustworthy." International Journal of Human–Computer Interaction 36.6 (2020): 495-504.

**Support:**

3

---

> ### Author Rebuttal · Authors · 2026-03-30
>
> We thank Reviewer hN3U for the constructive feedback and for recognizing our clear position, interdisciplinary breadth, and up-to-date references.
>
> **W1: Schema is informal.**
> We appreciate this observation and want to clarify a deliberate design choice. Our schema is intentionally qualitative and intended as a first-pass structuring tool for interdisciplinary groups---not as a quantitative risk-scoring instrument. We argue this is the appropriate level of formalization for a position paper that targets researchers across disciplines (CS, law, ethics, economics), many of whom would not share a common formal language for risk quantification. Imposing numerical scales prematurely would risk false precision and obscure the very tensions we aim to surface. That said, we fully agree that future empirical work should develop operationalized versions of the schema. We note that established frameworks such as the NIST AI RMF similarly begin with qualitative risk tiers before operationalizing them---our schema occupies an analogous early-stage role, specifically tailored to the PI threat landscape. We will make this clear in a revised version of the paper.
>
> **W2: The interdisciplinary call is not novel.**
> We respectfully note that both works are already cited in our paper (Sections 2.3 and 1) and directly inform our framing of PIs as a socio-technical phenomenon. We build on Selbst et al.'s argument against "fairness and abstraction" by showing how it concretely manifests in the PI context --- where the abstraction trap is particularly acute because PIs blur the boundary between data and instructions in ways that purely technical defenses cannot resolve. Similarly, Shneiderman's call for human-centered AI motivates our automation dimension and the graduated human-oversight recommendations across risk levels. Our contribution is not to repeat the general interdisciplinary call but to instantiate it for PIs specifically: we provide a concrete schema, a stakeholder taxonomy adapted from OECD/EU AI Act terminology, and differentiated recommendations per risk level — none of which exist in prior work for the PI domain. We will strengthen the explicit connections to [1] and [2] in the revised manuscript to clarify this relationship.
>
> **W3: Engagement with AI governance literature.**
> This is a fair point, thank you. We already engage with the EU AI Act and the EU Digital Services Act, and discuss liability allocation across stakeholders. We will expand our governance discussion to explicitly reference the NIST AI RMF (particularly its govern-and- map functions, which align with our stakeholder-focused approach) and recent AI liability scholarship (e.g., the EU AI Liability Directive proposal). We note that our Case #3 already touches on financial sector regulation, but we agree that connecting it more systematically to the broader AI governance landscape would strengthen the paper, which we will do in a revised version.
>
> **Q1: Schema validation.** We developed the schema as an interdisciplinary group of scholars in computer science, cybersecurity, philosophy & ethics, law, and economics---precisely the kind of team we advocate for. The case analyses in Section 3.3 reflect our joint deliberation process. This framework is intended to serve as the foundation for empirical work across a wide range of real-world use cases from the community, which will not only provide external validation of the schema itself but also address gaps, such as the need for more reference cases. Broad validation with a wide variety of real-world use cases requires community involvement, which we aim to kickstart with this position paper.
>
> **Q2: Schema generalization to the other LLM safety risks.**
> In its current, relatively abstract form (see W1), we indeed do think that there is potential for such a generalization, though our focus is on prompt injections. We picked this focus deliberately to do justice to the limited length of a position paper. But it is clearly true that other types of LLM safety risks, such as jailbreaks and hallucinations, also elude mere technical mitigations and would thus benefit from a multidimensional interdisciplinary analysis. If the dimensions we suggest for PIs will transfer directly or if they might have to be adapted or just explicated with different concrete scales, remains to be investigated when studying other LLM risks in more detail.
>
> **Q3: Can the framework adapt to injectors who also adapt to governance measures?**
> This is an excellent point. Our framework is designed to be applied iteratively: as the threat landscape evolves, cases should be re-assessed along the three dimensions. The arms-race dynamic the reviewer identifies is precisely why we argue against purely technical solutions and for ongoing interdisciplinary engagement. Adaptive injectors may shift a case from low to higher risk, triggering different mitigation strategies per our graduated recommendations in Section 4.

---

> > ### Author Rebuttal · Reviewer_hN3U · 2026-04-04
> >
> > Thanks for the author's response. I choose to maintain my initial rating.

---

### Decision · Program_Chairs · 2026-04-30

**Decision:**

Reject

**Comment:**

PI attacks (as most other AI security threats explored over the last several years) have been considered in the literature almost exclusively from a purely technical point of view---one of designing robust LLMs or defenses (e.g., detectors) in the context of such attacks.  The paper proposes a significant shift from this paradigm, considering the problem of PI threats holistically through a multidisciplinary lens which captures not just technical, but also social and legal dimensions of the prompt injection risks.  All reviewers greatly appreciated this interdisciplinary perspective, as well as the associated (qualitative) schema proposed by the authors.  The reviewers did raise a number of concerns, which they felt were adequately addressed in the rebuttals.  These include:

1) Interdisciplinary approaches are a common refrain in modern AI research, with such approaches called for across a broad array of related areas, such as fairness. As such, the position ultimately has limited novelty in this broader perspective.  However, as the authors argue, this approach has *not* been sufficiently explored in the context of PI risk, where the dominant perspective is purely technical.

2) The paper is extremely euro-centric, using EU regulation as its primary examples.  While the addition of a discussion of US regulatory work (e.g., by NIST) will broaden scope, it ultimately lacks a cross-cultural perspective.  The authors' response promises to remedy some of these issues (such as adding a discussion of NIST AI RMF), but I feel that they did not take the lack of a multi-cultural perspective sufficiently seriously.  Even though the authors argue that regulation elsewhere often follows EU regulation, this is not the same as cultural understanding and sensitivity, particularly outside the distinctly European context and influence.

3) There is the issue of how to structure interdisciplinary collaboration.  While the paper offers some ideas, it seems to stress a particular mix of fields: CS, law, ethics, economics.  Whey these, and only these?  What about political science, international relations, sociology?  Arguably even more importantly, within each is a broad array of subfields and perspectives.  Economics involves labor economics, micro, macro, financial, game theory, etc.  So, which economics?  Ethics has an enormously broad array of perspectives in philosophy, psychology, and cultural anthropology.  Which disciplines and perspectives do we include?  By definition, other disciplines and perspectives are excluded.  This is a very non-trivial issue that is hidden under the facially appealing notion of "interdisciplinary", and incidentally, does not even get into the fact that people often have strong disagreements within a discipline (particularly in the context of ethics).